# Challenges in accessing health care and socio-protection services among children living and working in streets in northwestern Tanzania: A qualitative study

Lilian Solile[1], Elias C. Nyanza[1]*, Joseph R. Mwanga[2], Dorice L. Shangali[3]

1 Department of Environmental, Occupational Health and GIS, School of Public Health, Catholic University of Health and Allied Sciences, Mwanza, Tanzania, 2 Department of Epidemiology, Behavioral Sciences and Biostatistics, School of Public Health, Catholic University of Health and Allied Sciences, Mwanza, Tanzania, 3 Department of Internal Medicine, Bugando Medical Centre, Mwanza, Tanzania

* elcnyanza@gmail.com

**Data Availability Statement:** All relevant data are within the paper and its Supporting Information files.

## Abstract

The escalating number of Children Living and Working in Streets (CLWS) in Tanzania has become one of the neglected Public Health issues. It is of more concern that, most of the CLWS hardly have access to health care and socio-protection services as a result, increase their vulnerability to infections and engagement in risky behaviors such as early unprotected sex. Currently, efforts by Civil Society Organizations (CSOs) to work with and assist CLWS in Tanzania are promising. To explore the role of CSOs, preventing barriers and existing opportunities in enhancing the access to health care and socio-protection services among CLWS in Mwanza city, northwestern Tanzania. A phenomenological approach was used to explore a full understanding of the individual, organizational, and social context factors on the role, prevailing barriers, and opportunities CSOs play in enhancing access to health care services and socio-protection among CLWS. Majority of CLWS were males, rape was commonly reported among CLWS. Individual CSOs are involved in resources mobilization, provision of basic life skills, education on self-protection, and mobilization of health care services to CLWS who depend on donations from public passersby. Some CSOs went as far as to develop community-based initiatives that give CLWS and home-bound children, access to health care and protection services. Older CLWS sometimes compromise the young ones from getting proper health care services by taking and/or sharing medication prescribed to them. This could be leading to incomplete dosing when ill. Moreover, health care workers were reported to have negative attitudes towards CLWS. Limited access to health and social protection services put CLWS lives at risk, calling for immediate intervention. Self-medication and incomplete dosage are a norm among this marginalized and unprotected population. Individual Civil Society Organizations attempt to address the needs of CLWS with a lot of barriers from the community and the health care system. It is time for the CSOs attempting to assist the CLWS to get support from the authorities and other people to aid this vulnerable population.

**Funding:** The authors received no specific funding for this work.

**Competing interests:** The authors have declared that no competing interests exist.

## 1. Introduction

Over decades Civil Society Organizations (CSO) have become more important in assisting various governments globally in strengthening the current health care delivery systems and socio-protection programs among the street children [1]. The rationale for CSO's involvement is linked to the development of a paradigm advocated by United Nations (UN) agencies and other leading development partners, which is increasingly adopted by many African countries including Tanzania [2]. African CSOs have increased in number over the past decade as a result of this dedication [3]. Most of the key interventions provided are focusing on social protection programs, illness prevention and health promotion in addition to treatment among Children Living and Working in Streets (CLWS) [4].

The CLWS are individual children under the age of 18 who spend most of their lives working for pay or begging for money from passersby in streets [4]. It is estimated that there are up to 150 million CLWS in the world, 2.8% of the total children world population [4]. These children are more likely to partake in high-risk activities like early sex, have concurrent partners, unprotected intercourse, illegal drug usage, using shared needles when self-medicating, group sex, sex trade, hence falling sick from Sexually Transmitted Diseases (STDs) and other infectious diseases like human immunodeficiency virus, acquired immunodeficiency syndrome (HIV/AIDS) [4, 5]. Additionally, multiple investigations revealed that several CLWS had little to no contact with their family or guardians and that the majority of them resided in risky areas of the city, such as under bridges, near marketplaces, by lakes or ocean sides, and other dim areas [6, 7].

More recent headcount surveys conducted in Tanzania show a total of 6,393 CLWS aged 0–18, were counted in the six cities, 51% of the CLWS aged 15–18 [8]. While the most common nighttime activity for male CLWS was sleeping (39%), an overwhelming proportion of females engaged in sex work (79%) [8].

Numerous studies have been conducted worldwide to investigate the health profile of CLWS, ranging from affordability to accessibility of heath care services [9]. Studies have revealed that the CLWS have limited access to health care services or no access at all due to high hospitalization and that consultation expenses in healthcare facilities [10, 11], is a significant barrier for CLWS who make little to no money on the streets. Other barriers included stigmatization by health care providers, minority status and not being sure of the quality of care they will receive in health care facilities due to their disadvantaged status [12].

Arguably other studies show that some CLWS could not find time to visit health care facilities as they struggle during the day to raise money for food and necessities and are only free at night [13]. This proves the needs of more studies to yield further evidence on the accessibility of health care services amongst the CLWS. Sexually transmitted infections (STIs) like gonorrhea and HIV/AIDS among CLWS have been reported to be very high, some studies showed that it can be higher than that for female sex workers, truck drivers and prisoners [14].

Currently, there are two common approaches to address the problem of street children in Tanzania. Institutional approach has been the most commonly practiced, to establish permanent institutions to provide shelter to the street children [15]. This approach tends to see the street children as wrongdoers, who need to be removed from the streets. The second approach is the Centre-based one. This assumes that it is possible and needed to remove the street children from their street environment for at least a short period [16]. Individual children may make use of these centers purely on a voluntary basis. So much will depend on and be determined by the Centre's ability to meet the needs of these children. Non-Governmental

Organizations (NGOs) run most of these centers. The most known names for these centers are "*Dogodogo*" in Dar- Es- Salaam, "*Kuleana*" in Mwanza, and "*Mkombozi*" in Moshi-Arusha and so many other centers in Tanzania. The centers may address basic health and education needs, provide further education or vocational training and offer opportunities for the street children to express their needs and fears for them to receive helpful support and advice [17]. There have been some critics to these Centers' operations as their operations are limited to children with more understanding, those who are mature and may not be accessible to other children, who perhaps needed more help [18].

Generally nutritional disorders, physical injuries, parasitic, and other community-acquired infectious diseases (HIV, malaria and other STDs), sexual and reproductive health disorders, violence and sexual abuse, substance use, mental health problems, transactional sex are all threats to CLWS [19, 20]. The fact that socio-protection services and access to health care remain negated is even more concerning. A very recent survey in Mwanza, reported a HIV prevalence of 12.2% among CLWS. The provision of health services and health care among CLWS is low and/or inadequate, resulting in disparity in access to health care services even though CLWS are exposed to difficult health situations and are more vulnerable. Recently, more promising efforts by CSOs to work with and assist CLWS in Tanzania has been reported [4]. However, the roles, barriers, and opportunities of the CSOs in addressing the CLWSs health and social security challenge remains unanswered. Therefore, this study aimed to examine the critical role played by the CSO's in strengthening health care services and social protection support among CLWS. The study further explores barriers and factors that if well addressed and/or integrated, could enhance access to health care and socio-protection services among CLWS in Mwanza city. The study was guided by a conceptual framework on Health care services among CLWS developed by the authors using different covariates from multiple studies published elsewhere [19, 20]. The conceptual framework included different factors that influence generation of CLWS, factors that hinder access and prevailing opportunities in enhancing access to health care and socio-protection services among individual CLWS to health care and socio-protection services to CLWS are summarized in Fig 1 below.

## 2. Methods

### 2.1. Study design, population, and settings

This study employed a phenomenological approach [21], cemented by qualitative approaches (i.e., *Key Informant Interviews (KIIs), and Focus Groups Discussions (FGDs)*) to fully understand role of individual, organizational, prevailing barriers, and opportunities of organizations in enhancing access of health care services and socio-protection among CLWS in Mwanza city, Nyamagana District northwest Tanzania. The city is reported to have around 39.6% of the entire population as young population. Children have been reported to migrate from rural to urban areas for ecological and historical reasons [4]. Social services in rural areas are inadequate and consequently these areas could not attract people and they had to migrate to urban areas to search for work [4]. It is on this basis, Mwanza city qualified as a study area. Mwanza city has 34 public health facilities which includes 2 hospitals, 5 health centers, and 27 dispensaries.

A systematic random sampling procedure technique was applied to select primary health care facilities where health care workers participating in the study could be located [22]. A purposive sampling procedure was used to recruit CSO involved with CLWS [23], Nyamagana District social welfare department staff, and health care workers who were involved in the provision of health care and socio-protection services among CLWS. For a sub-group of CLWS, a

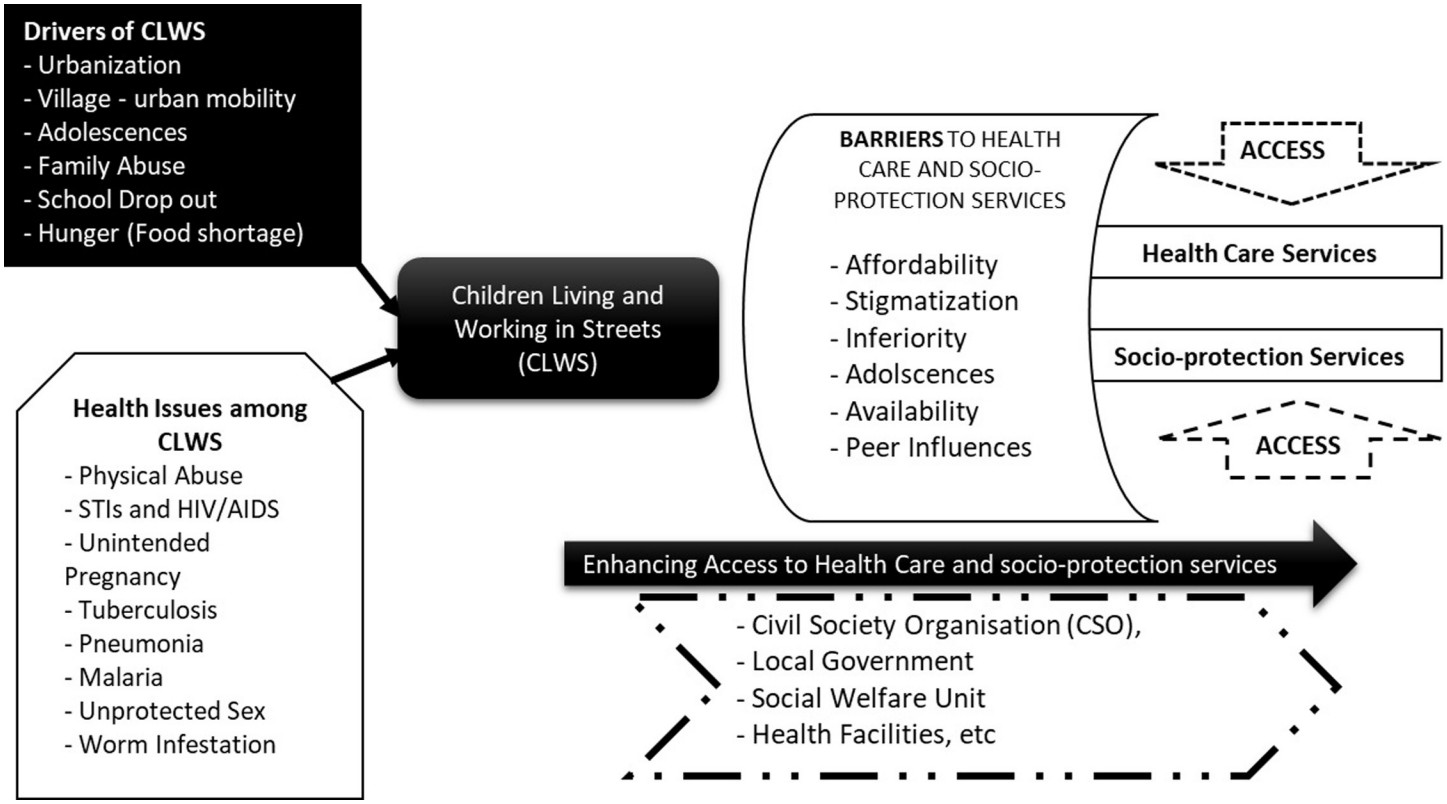

**Fig 1. Conceptual framework on health care and socio-protection services among CLWS as summarized from different studies elsewhere [19, 20].**

convenient sampling technique was used with the guide and assistance from the Nyamagana District Social welfare department [24].

The study was divided into three phases: $1^{st}$–Identifying specific CSO involved in supporting and strengthening access to health care services and socio-protection among children living and working in different streets in Mwanza city. $2^{nd}$–Inviting individuals working with CSO, and Nyamagana District Social Welfare department to a face-to-face interview (i.e., *through KIIs and FGDs*) on issues related to the support and strengthening access to health care services and socio-protection among CLWS in different streets in Mwanza city. $3^{rd}$–Inviting individual CLWS to a face-to-face interview regarding the types and kinds of support they get in accessing health care services and socio-protection in Mwanza city.

## 2.2. Data collection

In observing the existing phenomena and practices, we collected data using three different methods to triangulate sources of information as follows; $1^{st}$—One interview with CSO and social welfare department Key Informants at Nyamagana District (i.e., *KIIs*) who are involved in support and strengthening access to health care and socio-protection services among CLWDs in different streets of Mwanza city. Therefore, using purposively sampling, at least 14 KIIs were conducted until thematic saturation was attained. Each KII took at least 30 to 45 minutes. $2^{nd}$ *FGDs* with members of the CSOs, Nyamagana District social welfare department and sub-group of children living and working in different streets in Mwanza city were also conducted. A total of 12 FGDs were conducted to saturation of emerging themes. Each FGD took at least 45min to 1 hour.

## 2.3. Data analysis

Thematic content analysis was used to categorize emerging themes after every interview [25]. We commenced with one interview, transcribe verbatim, code and analyze for emerging themes and review the questions to see if the right question is asked [26], and continue with another interview until no new theme is emerging (maximum saturation). Additional interviews were conducted as needed to pursue relevant themes as they emerge, with a goal of theoretical saturation. The audio files from KIIs and FGDs were transcribed verbatim and later translated from Swahili to English and then back to Swahili to ensure we have common understanding. Data obtained at every interview was used to guide coding rather than imposing a coding scheme. Coding was done by two people to increase investigator triangulation [27]. Finding themes from data was done and then documented and finally emerging themes were analyzed, and inferences made. NVivo 12 computer software—QSR International 20—was used to enhance data analysis [28].

**Table 1. General characteristics of the study participants.**

| Variable | Category | N | % |
|---|---|---|---|
| **For Children Living and Working in Streets** | | | |
| Sex | Male | 14 | 63.6 |
| | Female | 8 | 36.4 |
| Age | 9–12 | 5 | 22.7 |
| | 13–15 | 8 | 36.4 |
| | 16–18 | 9 | 40.9 |
| Education | No Formal Education | 14 | 63.6 |
| | School Drop-Out | 3 | 10.3 |
| | Primary School | 5 | 22.7 |
| Street Activities Involved | Collecting Bottles | 14 | 63.6 |
| | Sweeping Public Buses | 5 | 22.7 |
| | Fishing (Hooking) | 3 | 10.3 |
| Been Sick | Yes | 18 | 81.8 |
| | No | 5 | 22.7 |
| Ever Treated at Health facility | Yes | 14 | 63.6 |
| | No | 8 | 36.4 |
| Ever been abused | Yes | 22 | 100.0 |
| | No | - | - |
| Ever been sexually abused | Yes | 13 | 59.1 |
| | No | 9 | 40.9 |
| **For Members from CSO** | | | |
| Sex | Male | 18 | 37.9 |
| | Female | 11 | 62.1 |
| Age | 25–30 | 3 | 10.3 |
| | 31–35 | 4 | 13.8 |
| | 35–40 | 8 | 27.6 |
| | >40 | 14 | 48.3 |
| Education | Primary School | 3 | 10.3 |
| | Secondary School | 10 | 34.5 |
| | College and above | 16 | 55.2 |

### 2.4. Ethical consideration

Ethical approval for the present study was granted from the Joint Catholic University of Health and Allied Sciences and Bugando Medical Centre Research and Ethics Review committee (*Ref.*, *CREC/594/2022*). Then, permission to conduct this study was requested from Regional administrative office (*Ref.*, *No. GB.47/333/03*), City Council executive office, and from selected CSO in Mwanza. Permission was obtained from the respective CSOs, Nyamagana District social welfare department, respective health care facilities followed by written consents among individuals selected from such organizations and institutions before being invited to participate in the face-to-face interviews (KIIs) and/or FGDs. Since individual CLWS are under the age of 18, efforts were made to ensure they are protected. Individual CLWS had to fill assent forms to show their agreement and willingness to participate before being allowed to participate in the study. In instances where participants had low literacy or were apprehensive about signing formal documents–efforts were made to read and explain about the scope of the study. All participants were informed that they could withdraw from the study at any time without impacting the care and/or services they receive.

Prior to obtaining consent and assent from study informants, a detailed explanation of the purpose and procedures of the study and why the participant was selected was done. Participants were also assured of strict confidential management of the data, and that, the information they provide would remain anonymous. Identification numbers were used in the study instead of names. The study informants were issued with a copy of the signed informed consent/assent form while the researcher retained the other copy.

## 3. Results

### 3.1. General characteristics of the study participants

Information to enable answering the study objectives were collected from individuals working with CSOs (N = 29) (Four CSOs were recruited), individuals working with health facilities that were responsible for provision of health care services to CLWS (Two Public Health Facilities and One Private health facility, and a sub-group of CLWS (N = 22). The distribution of their demographic characteristics are summarized in Table 1.

### 3.2. Identified themes in enhancing access to health care and socio-protection services among civil society organizations to CLWS

Four key themes and one sub-theme emerged from interviews, FGDs, and a review of CSOs' documentation, all of which revolved around the need for support to improve access to healthcare and social protection services among CSOs to CLWS.

**Theme 1: Systems support.** The study showed that several organizations responsible for provision of health services, social protection to CLWS are available as acknowledged by a worker in one social welfare department;

*"Yes, there are many organizations, the biggest organization is Railway Children, while Cheka Sana is another organization that provides aid to street children. Both organizations help to take injured children to hospital and take them back to their premises in collaboration with CF Hospital. Other small organizations are Farijika, Fanisi, Rafiki for children and Upendo Daima which have centers that provide shelter for the CLWS"* [WO1]

Such organizations provide a wide range of services to CLWS and their families such as education on basic human rights and supporting health systems. As one participant from Fanisi Organization described that,

*"Fanisi is a Non-Governmental Organization (NGO) founded in 2019 and originates from Mwanza, Ilemela district. Also, Fanisi is community based and helps families"* [FOM1]

Another participant from Railway Children organization added that,

*"Railway children deals directly with street children living in harsh environments. The headquarters are in Dar es Salaam, we also operate in Mwanza and Dodoma in partnership with TIZEBET organization, to provide funds to enable our organization to work effectively for CLWS"* [RCF]

He insisted that *"We have four major activities; starting with street outreach where we visit children in various streets of Mwanza. These children are mostly found in Ilemela and Nyamagana districts at bus bays or in densely populated streets. Most of the children are in Nyegezi streetwhich has a bus bay Natta, Mabatini and Muslim cemeteriest, where we get to talk to them. . . Nyamagana district links with Kamanga and the city center on the ferries from Sengerema, children leave from Bukoba to Kamanga, and some prefer to stay there since there is a lake where swimming is free. Igogo, Kirumba ward and Mwaloni markets in Mwanza are places with a lot of drunkards, so we always work and go to those places and get information from the children, stakeholders, and the government itself. We visit these children at different times of the day. Here we get to know them and make them trust and work with us. This makes it easy for us to understand their situations. In case we find sick children, we take them to hospital and for children who just arrived in streets we make speedy efforts to take them back to their homes".* [RCF]

He added, *"The government also helps us with the children's safety, in terms of shelters. . .but also during our outreaches we get support from community members, good Samaritans, food vendors, who have been of great help for the children since we get feedback from the children about the people who helped them, we call them our "community champions". We talk to these people and give them training, we advise them to take the children to the social welfare officers, however people do not follow the law, and they take the children and live with them, which is very dangerous. Sometimes we receive calls that the rangers are beating the street children, so we get there with the police and take them to the social welfare officers and they provide shelters for them."* [RCF]

It was mentioned that CSOs support street children with some basic needs as one respondent from an organization described,

*"Our main responsibilities are to donate open shoes and bites like biscuits, our institution also provides education, this was established by one of the former doctors in Mwanza. We also provide all the health services to the street children for free but protecting them from cruel acts is difficult as we don't operate 24/7".* [CFF]

*"We also provide nutritional support for children with malnutrition with the help of doctors."* [FOF1]

Moreover, welfare officers and organizations help children get back to their families as he described

*"In addition, after talking to CLWS, if they show willingness to go back home and the correct details of their families are found, children are returned to their families after being sent to health centers for checkup to ensure their families' safety."* [WO2]*"*

*"When we return those children, it is the responsibility of the Social Welfare Officer to continue educating parents on child abuse and child labor in order to abolish it".* [WO1]

*"We also communicate with social welfare officers in various regions of the country to get information on children's' home. For Example, a child says that he came from Ngara district, the responsibility of the social welfare officer in Ngara is to find out the family's information, reasons for the child's escape, and the health status of the child by filling out a form, then after we give permission for a child to stay in a shelter or reunite with their families."* [WO1]

*The second intervention is family integration, also called family rule reification. (Reification is to find a child and return them back to their home). We have children from different parts of the country. Some appear to come due to loss of parents, poverty, adventure, and some peer pressure. Because of similar reasons for residing in the streets, we use different techniques to re-unite them with their families. In the past, we used to think that if the child comes from Kigoma, we would take him back to Kigoma, this was time consuming, the children barely giave reliable information, so we decided to do network tracing with the officers in the area the child mentioned until we got the correct place where the child lived with the help of the Chairman or the Executive.*

*Once we find the child's family, we do a household assessment to find out the family health, relationships and economic state and the issues related to sexual violence. We do research to know the sources of leaving home to make sure that if they are returned to their homes they won't go back to the streets.*

*Families who are self-sufficient, we tend to provide capital for business support with the help of development officers so that they can provide for the needs of their family. We provide CHF insurance which we believe after one year a family can pay for that insurance. So, it depends on the needs of the family, and it's not that we shall provide each and everything, we always talk to stakeholders or the government, they provide equipment, but not to a larger extent, sometimes organizations help us."* [RCF].

*"We also work with young people in the streets whose priority is not to return home, we talk to them, we empower them. We connect them to vocation training center so they can get knowledge on self-dependency and stop engaging with illegal activities in streets"* [RCF].

However, children returned home sometimes go back to the street as officers said,

*"Challenges with the street children, is that sometimes they are not on a common goal, some of them have been taken back to their parents but after few months they return in streets. They tend to make fun of us especially if they meet you"* [WO1].

Organization get support from the government as one participant from Railway children said,

*"We have been working closely with the DMO office, hospitals, and social welfare officers on special days like AIDS Day, Youth Day where they give us chance to get street children involved. In case of various opportunities, they communicate. For example, the executives gave information about vaccinations for children, so we take our children for vaccination.*

*Also, in case of provision of needs and equipment we get information. Also, during COVID19 pandemic, children were given sanitizers, masks, and soap".* [RCF]

**Theme 2: Gender issues.** Participant described that the number of boys and girls in the street is not the same, as one staff from Railway children described that,

*"The number of female children in streets is lower than male children. A girl child is marketable in streets they can work in local restaurants, or as house maids." [RCF]*

*"In cases of sexual violence, it's hard to find the accuser, since they barely remember what happened to them. When a child is raped, we take her to the hospital for treatment. However, it is very difficult for street children to go to the hospital and get help. Another challenge is trust, the society doesn't trust children even if you educate them, some people tend to be cruel to them, they are given bad names, sometimes they get beaten"* [RCF]

*"We have had many reports on male children, and these have led to a sub project that is related to cases of sodomy, although there are cases of female children, but they are not as many as the male children."* [FOM2]

It was also revealed that CLWS are protected from sexual violence.

*"There are many opportunities provided by the local government and the Community Welfare Officers, we provide a permit for free treatment in the hospital for all victims of sexual violence, they can also have access to maternity tests through the support of the government or organizations."* [RCF]

*"Children need to be befriended and visited frequently by social workers for them to gain trust in them and open-up. If you meet the child on the first day, he may not open up, but you may need up to a month for them to open up. The good part is that the organizations tend to have employees who are in the streets every day, they are close to the children, and they prepare and bring them to us to get them the basic needs that we offer."* [FO1]

*"We provide education to the children against violence. We often advise them not to sleep in certain places alone, not to be in dark places, not to receive things from strangers. We always escort the children to social workers since some of them are not confident and can't easily express themselves. We provide education to adolescent children about sexual reproduction and provide training and HIV/AIDS tests."* [RCF]

**Theme 3: Economic support.** Organizations have donors that support them financially for them to be able to support street children.

*"We have had the opportunity to get funds from donors who rely on to complete the goals of the organization." [FOM2]*

*"We have very few donors compared to those in need in our communities, so there is great need for our management to find donors so that we can reach our organizational goal."* [FOF1]

In addition, parents are involved in one way or the other to support them as it was explained,

*". . .we encourage parents to create the financial supportive groups among themselves so as to raise and support their children well."* [FOM1].

*"In the community we work with guardians at the level of wards where we have created clubs, each club has 60 members who meet every month for training. Another approach which is less costly due to our shortage of funds, we plan to meet mothers at clinics to educate them on how they will be able to create positive relationships with their children."* [FOM1].

*Sub theme*: *Supportive health services*. Organizations are aware that street children face different health issues, and they are supporting them as one described that,

*". . .we have a lot of needy people like 1400 children with health needs who are all vulnerable. And we are focused on helping those children, but we are trying to find local donors, organizations that are focused on the health sector to save those children."* [FOM2]

*"We don't have a clear number of those that we have helped to access health services, however if they go to the hospital without money, they cannot provide him with treatment until we give them exemption documents from social Welfare officers. We have agreed that if any child gets sick from the streets, we take him to the AIC Church Hospital, and mission areas. We leave the medicine under the care of a local food vendor for dose management. If a child is raped, we take her to Hope Child Center or Sekou Tuore (RRH), it's easy since it's cheaper, and we provided NHIF cards for about six children who have been on the streets for a long time."* [RCF]

FO and RC assist the community in managing different systems such as health insurances to children living in the streets as one participant said,

*". . .there are many organizations, including the MOSCO organization, which provides health insurance to children. We decided to prevent increase of street children through family interventions which focuses on providing therapy sessions for children and parents about the problems they have encountered in their lives, community intervention which provide positive sessions for parents, we also have clubs that we visit every month to educate parents about positive education and upbringing better to prevent child abuse."* [FOF1]

Furthermore, it was found that these organizations help families with street children get health insurance or medical help whenever needed as explained by one of CSOs worker:

*"At Fanisi have a procedure of working with the local government leaders, so the family will be recognized by the local Government leaders. So, once the chairman brings us the number of families who live in a harsh environment, we usually find families that need health insurance and in case they experience challenges we usually accompany them to Bugando Hospital, we usually pay the expenses ourselves. We also cooperate with other organizations such as Forever Angels, Fonelisco, who provide shelter for CWLS."* [FOF1]

Another social welfare officer added that they usually communicate with social welfare department of different health facilities to ensure that CWLS are provided with health services,

*"We are a health department; we don't find it difficult at every health center or hospital. If there is a problem, we contact the welfare officer at a certain hospital and the child is given an exemption for treatment."* [WO2]

Furthermore, it was revealed that police are involved in cases of cruelty to ensure children's safety though it's a challenge when they escape.

*"We have our IDs for street children and the hospital recognizes our IDs, so if a child goes with that ID, he must receive treatment. The hospitals that provides the most aid is Sekou-Toure, Butimba Hospitals, Nyamagana, Makongoro and Buzuruga. We have social welfare officers there hence we easily get treatment. But we also recognize the issue of cruelty, we provide trainings at the health facilities so that when there is a case, we can have access to PF3 form and with the cases in court, we make sure we fill in the records."* [RCF]

*"The biggest challenge is for cases that go to court, to get a doctor to write PF3 form for you is sometimes difficult, others tend to refuse to fill in those forms, but we get good cooperation sometimes"* [RCF]

*"Another challenge is when children are taken to the court, they give you guarantors, but we don't have a center for victims, you must ask them to stay with someone or return to their own streets, the law requires the local Chairman to make sure the child is there. If the child escapes, the court demands you must find him."* [WO2]

However, these children sometimes are involved in crimes like theft to people who help them as welfare officers explained,

*"We have received a complaint that when these children are admitted to the ward, they tend to steal from other patients' belongings, so we decided to separate them from other patients but once they are done with the treatment they leave, another challenge is improper dose usage, it leads to the child's body weakness, the children report that the medicines are taken by the older boys."* [CFF]

**Theme 4: Provision of education.** Interview revealed that street children are supported in their education as one participant responded,

*"The method we use to reach three quarters of the children is through institutions. In Mwanza Region, many children are in schools, so we thought it is the best way to provide them with education, we have commenced with few schools, 30 schools to be specific, we have started something called a class club in schools, which is against child abuse. In each school, there are at least 20 members in each club, we work hand in hand with teachers. We have noticed a change, the children now recognize any forms of child abuse, and once they experience, they tend to report to the club leaders and the club leaders inform the teachers and the teachers convey the information to us."* [FOF1]

*"We are working with CUHAS, the social welfare and community development officers found in every hospital, Buzuruga Hospital, BEIRA, and clinics, all these work with us in educational provision for the parents and guardians."* [FOM1]

It was also discovered that through CSOs children who dropped out of school went back to school, as evidenced by one speaker:

*". . .there is good progress concerning the children who had given up with school, and now are back and are progressing well academically."* [FOM1]

The communities surrounded with street children are educated on how to live with them and protect them from violence.

*"Educating the community on how to help children, cooperate with the police and the social welfare officers in hand with the hospital on our services"* [RCF]

*Sub theme*: *Human rights education*. It was revealed that society still needs to be educated about children's rights and how to overcome stigmatization.

*"At Fanisi, we are currently implementing two projects, where we have a parenting project, (the Malezi Mothers Project). This project is implemented in three districts which are; Sengerema District, Nyamagana District, and Ilemela District. We also have another project that is being implemented in the Sengerema district alone. These projects make the child, and the parents realize their basic rights."* [FOM2]

*"There is no transparency from the community, for example if a child is brutalized, there is little cooperation from the child's family in identifying the person who brutalized the child, so the incident is hidden, they know that they are protecting the honor of the family, but we at Fanisi realize that the child is affected and can be harmed, if the person responsible is identified, we take the person responsible/perpetrator of the crime in the relevant area and the law continues to act."* [FOM1]

*"We rescue the children, we provide them with medical help, if the child has been brutalized, we provide legal aid and we also protect them, we strive to ensure that the children's rights are respected. We also connect these children with their families, these are some of the activities we do in the organization."* [RCF]

## 4. Discussion

### 4.1. General overview

Several CSOs were found providing some basic social services and social protection among CLWS around Mwanza city. Indicating prevailing social initiatives from both the CSOs and the governmental social welfare departments. Individual CLWS do live and stay in groups in some specific locations (*namely*, Nyegezi, Pamba Rd, Mabatini, Muslim cemeteries, Kamanga Ferry, Igogo, Kirumba market, and Mwaloni area) within the city. The number of CLWS is not clear even though the great size remains high it is not well documented. One CSO reported attending a total of 1400 CLWS in a year by provision of food, medical care and re-connecting with their families.

Gender inequality emerged among CLWS where majority of the present children were male. It was learnt that, among the CLWS individual female children get employed easily as compared to their counterpart male children. It is important however, to make a follow up on the labor market among such girls as there were no records of where they work, to whom do they report, and how are they treated. Our findings are consistent with a previous quantitative study where the percentage of male was dominant among CLWS as compared to female (70.6% vs 29.4% [20]. This is also supported by studies from three cities in Cameroon where the number of female CLWS were 19.8% whereas their male counterparts were 80.2% [21].

It was also reported that stranger kids arriving from their local villages are subject to victimization and harassment from individual children or from the surrounding and interacting

communities. This is a vital alert as they may be victims of assaults including sexual abuse from individual children or from the surroundings and interacting communities. In a subgroup of CLWS, 59.1% reported to have been sexually abused. Incidences of rape among CLWS are said to be common and have been reported from other places in India, Sudan, South Africa, and Cameroon [21–23].

Our findings indicated that, individual CLWS are subject to environmental stress associated with weather risks and lack of appropriate and reliable shelter. This forced them to use ditches and trenches, subjecting them to potential infections and other health challenges. Also, previous studies have reported such children sleeping in half-destroyed houses, abandoned basements, under traffic bridges and in the open suburbs–a homeless lifestyle [19]. These findings are like those reported among street children in N'Djamena Chad, South Africa, Morocco, Peru, and Pakistan [21, 23–25]. It is also important to examine current environmental exposure health related problems as CLWS mentioned to be sleeping in half-destroyed houses, abandoned basements, under traffic bridges, trenches, and ditches.

In a discussion with a subgroup of CLWS, 63.6% reported to be engaged with collecting plastic bottles where 22.7% were engaged in sweeping public buses at their respective boarding areas. It is important to note that, such activities are conducted with bare feet, and without any protective gears. This could be increasing their risk to infection including the novel Corona Virus (COVID-19).

It also emerged that individual CLWS are sexually abused so often, and it is always hard to find the abuser. Some of the CLWS are so exposed to sexual abuse such that, they get used to it for money or some bribe incentives from the surrounding adult men and women. This marginalized group is also exposed to homeless adults such as disabled individuals as well as drug addicts [23]. On this regard, it came out that, male children were more victims of sodomy as compared to their counterpart female children. Our findings are similar to those reported elsewhere, where high-risk behaviors such as early sexuality, concurrent sex partners, engaging in unprotected sex, drug use, sex worker, illness due to sexually transmitted diseases and other infectious diseases including HIV/AIDS are common among CLWS [4].

### 4.2. Role played by CSOs in enhancing access to health services

Regarding the roles played by individual CSOs in Mwanza city, it was revealed that the overall scope of the CSOs and social welfare departments around the city in enhancing access to health services and in soliciting social protection to CLWS are diverse. For instance, most of the CSOs, are involved in resources mobilization, provision of basic life skills and education on self-protection, human rights among CLWS, provision of shelter, food provision and mobilization of health care services such as medicine. The provision of medicine, however, is not based on medical laboratory diagnostic reports; and monitoring of dosing has always been a challenge. Also, it emerged that incomplete dosage for given medication was likely among CLWS as older children tend to take medication from the young ones. This could be contributing to drug resistance and/or escalating the public health challenges against drug resistance as declared by the World Health Organization [26]. This contention agrees with an analytical cross-sectional study in Mwanza city where gram-negative isolates and Salmonella species isolates were resistant to third generation cephalosporins [27]. Our findings are similar to those reported elsewhere in Chad and Cameroon, where self-medication has been reported creating some potential for drug resistance [21, 23]. Since there is a potential for self-medication and lack of dosage treatment compliance, a follow up study is therefore needed to examine the drivers and possible interventions to stop this Public Health calamity of which the consequences are beyond CLWS and could be affecting those in their environmental vicinity.

Other activities included but not limited to enhancing re-unification of CLWS with their families, provision of child support to families staying with the re-united child, provision of business support to children's family. Provide a one-time annual insurance plan which guarantees access to treatment to health facilities among individual families staying with the child. However not all individual children do get health insurance and some services may not be covered by such insurance. The issue of affordability has been reported in previous studies in different cities in Cameroon [21].

Individual CSOs are also engaged in provision of education to parents regarding some possible reasons for a child to escape, network tracing of respective children in efforts to identify their origin for possibly sending them back to their families. It also emerged that, CSOs are engaged in provision of sexual reproduction and HIV/AIDS education to CLWS. Since CLWS have been exposed to different environmental and social challenges, they need some special program before being integrated back into their respective communities. To address this concern, CSOs currently provide therapy sessions for CLWS and their parents regarding their problems encountered in their lives as rehabilitation sessions.

## 4.3. Potential barriers in enhancing access to health services and social protection to CLWS

The current findings indicated that individual CLWS are limited in accessing health services and lack social protection from those interacting with, and the community around. Even though there are efforts to bring individual CLWS back to their original homes, such efforts have not proven to be effective as most of the children return to the city. This could be caused by several factors; one being not addressing the root cause of the problem which may have contributed to the escape from their home communities to the city in the first place. It could also be attributed to the fact that most of the CLWS are getting used to street life and freedom of moving without being supervised. This could be associated with being so intoxicated with street life and the need not to be moved away.

This study revealed that underreporting of child violence from communities is the norm, as everyone wants to protect one another–protecting the abuser. Individual children can be brutalized by a member of a community and such incidences remain unreported. Even in the health care system, where to get a medical report when a child is brutalized and/or raped is not any easier and often impossible. This denies individual children justice as the abuser is not brought to justice and is likely to continue with such behaviour in the future. Incidences of rape among CLWS are said to be common and have been reported from other places in India, Sudan, South Africa, and Cameroon [21–23].

Also, community members who operate different businesses around the city among other Samaritans—identified as community champions in assisting CLWS—have been reported to be uncompliant with laws, regulations and child rights protecting children. For instance, some individuals have been reported to take such CLWS to stay to their homes without reporting or obtaining an approval from social welfare authorities within the city. There could also be some potential child assault in the umbrella of social protection among those who take the children to their homes that need to be investigated. In 2001, the International Labor Organization (ILO) in collaboration with the Tanzania Ministry of Labor reported a prevalence of 40% and 48% for economic activities and domestic child labor [28]. Most of the CLWS are reported to be engaged with scavenging, food services and food selling among others. Previously, in Mwanza city, 23.9% of CLWS were reported to be involved in cooking and food services for food vendors [20].

It was also revealed that, the CSOs do not stay or work for, or work on, even work with CLWS 24hrs a day or 7 days a week. This is one of the barriers in social protection for CLWS as in most cases such children are left roaming in the city during the night subjecting themselves to possible assault.

On the other hand, it was revealed that, CLWS are reported to be engaged in illegal activities such as robbery and sexual activities limiting the surrounding communities with a fear to be engaged in offering support, creating what is called social exclusion. It emerged that CLWS practice robbery even to people who attempt to provide some support. For instance, health care workers reported cases of theft of patients' properties when such children are admitted in hospitals. This among other things has led to limited access to proper health services among CLWS as health providers have a negative attitude towards them. Such negative attitude towards CLWS among health care workers has been reported among street children in Chad, Cameroon, and Bangladesh [21, 23]. It is important to understand that life on the street exposes children to a myriad of risks, omits their safety and comfort that a family environment can offer. Evaluation studies tracing medical services and care given to respective CLWS should be encouraged to understand the scope of medical care individual CLWS are given.

It was also revealed that CLWS do not provide full cooperation as they are used to and like to be in streets and do not support initiatives to send them back to their families. Also, the information provided, in most cases, is not correct and cannot be reliable. This could be one of the barriers and a driving force to the escalating number of CLWS in Mwanza city, however a follow up study is needed to confirm the same. Various studies in other parts of the world revealed that most CWLS are not ready to go back to their homes because they feel safe in streets than home, as the violence they experience in streets is lesser than that experienced in their homes.

Another potential barrier is donor dependency syndrome as most of the individual CSO operate their activities based solely on donor funds. Such CSOs lack self-generating sources of funds. This limits what they can offer and the number of those who can benefit from their activities. It is important to understand that, even though some of the CLWS get health insurance, the subscription provided does not cover high level medication and tends to cover basic medical care only, and not all CLWS get health insurance. This could limit those in need to access health care services. On the other hand, CLWS face some barriers in accessing health care services as health care workers refuse to sign their police required incidence reporting forms for individual children hence prolonging the waiting time the child would have to take before being attended to.

## 4.4. Opportunities

Even though CLWS lack direct social protection services; training provided by some of the CSOs provide essential basic skills and opens an avenue for learning how to avoid and seek help against sexual abuse among other harassment and torture from the surrounding communities and the public. Reported community champions within the city (namely; food vendors and good Samaritans) who assist children to get basic needs is another opportunity for reaching CLWS to enhance access to health services and in enhancing social protection.

The presence of hotspots as common areas for CLWS where they are found most often provides a platform for potential entry in provision of basic social services such as health education, health seeking education, and social life skills. Emphasis on education and health promotion have been suggested elsewhere [21]. Another opportunity which seems to be a mixed paradox, is the reliability of CSOs to national and international events, festivals, and commemorations where health services are offered for free. Individual CLWS in need of health

services are therefore mobilized to reach such events for possible medical care. This can be seen as an opportunity in one hand for accessing health services but also a barrier in the other hand. Relying solely on such events and exhibitions would be risking the health of those in need as they do not occur as often. It is also important overemphasizing on the reality that health challenges do not respect calendar days; can occur at any time.

There is also an emerging community-based opportunity for enhancing access to health services and creating social protection to the children at home and those working and living in the street from some of the CSOs. It was reported that CSO initiated financial supportive groups that aim at building capacity to communities so that they can take initiatives to protect their individual children in accessing health care and strengthening their social protection. It is important to understand that most of the rural-urban migration among CLWS is involved with poverty and need for secured livelihood [28]. It is also important to appreciate that CSO enhances better environment for children so that they do not feel the sense of being stressed and/or cohesive. Some CSOs provide training regarding realization of the rights of the child as well as the rights and obligations of the parents. It is also important to note that some CSO provide positive parental care education at the antenatal care clinics to enable parents to live in harmony with their children.

## 5. Conclusion

Individual Children living and working on the street face life-threatening limited access to health services and inadequate social protection that calls for immediate intervention. Self-medication and incomplete dosage are a norm among this marginalized, most excluded, and unprotected group. Individual CSOs attempt to address their needs with a lot of barriers from the community and health care system.

### Study strength

The use of phenomenological qualitative approach in the present study, enabled us to understand fully the individual, organizational, and social context factors regarding the role, prevailing barriers, and opportunities of CSOs in provision of health and social protection services to CLWS.

### Limitations of the study

In the study, we did not go across patient files to confirm that individual CLWS accessed health care services to the mentioned health facilities. Tracing medical records and care given to respective CLWS should therefore be given priority in future evaluation studies involving such vulnerable and marginalized groups. It was also difficult to access meeting minutes as well CSOs' evaluation reports, it seems that no evaluations are conducted. Another limitation in this study is failure to access records of brutal actions against CLWS.

### Supporting information

**S1 Data. Interview scripts.**
(ZIP)

### Acknowledgments

The authors acknowledge the Department of Environmental and Occupation Health at the Catholic University of Health and Allied Sciences for their support with this study. Authors

would like to thank all the participants of this study particularly CLWS, CSOs and health care workers.

## Author Contributions

**Conceptualization:** Elias C. Nyanza, Joseph R. Mwanga.

**Formal analysis:** Lilian Solile, Elias C. Nyanza, Joseph R. Mwanga, Dorice L. Shangali.

**Supervision:** Lilian Solile, Elias C. Nyanza.

**Validation:** Dorice L. Shangali.

**Visualization:** Joseph R. Mwanga.

**Writing – original draft:** Lilian Solile, Elias C. Nyanza, Joseph R. Mwanga.

**Writing – review & editing:** Elias C. Nyanza, Dorice L. Shangali.

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
