## [Decision Letter · Decision Letter 0]

27 Mar 2023

PGPH-D-23-00312

Challenges in accessing health care and socio-protection services among children living and working in streets in northwestern Tanzania: A qualitative study

Dear Nyanza,

Thank you for submitting your manuscript to PLOS Global Public Health. After careful consideration, we feel that it has merit but does not fully meet PLOS Global Public Health’s publication criteria as it currently stands. Therefore, we invite you to submit a revised version of the manuscript that addresses the points raised during the review process.

We look forward to receiving your revised manuscript.

Kind regards,

Collins Otieno Asweto, PhD

Academic Editor

Journal Requirements:

Reviewers' comments:

Reviewer's Responses to Questions

**Comments to the Author**

1. Does this manuscript meet PLOS Global Public Health’s publication criteria? Is the manuscript technically sound, and do the data support the conclusions? The manuscript must describe methodologically and ethically rigorous research with conclusions that are appropriately drawn based on the data presented.

Reviewer #1: Yes

Reviewer #2: Yes

Reviewer #3: Yes

Reviewer #4: Yes

2. Has the statistical analysis been performed appropriately and rigorously?

Reviewer #1: N/A

Reviewer #2: N/A

Reviewer #3: Yes

Reviewer #4: Yes

3. Have the authors made all data underlying the findings in their manuscript fully available (please refer to the Data Availability Statement at the start of the manuscript PDF file)?

Reviewer #1: Yes

Reviewer #2: Yes

Reviewer #3: Yes

Reviewer #4: Yes

4. Is the manuscript presented in an intelligible fashion and written in standard English?

Reviewer #1: Yes

Reviewer #2: Yes

Reviewer #3: No

Reviewer #4: Yes

5. Review Comments to the Author

Reviewer #1: This is an excellent manuscript which has illuminated well on a vulnerable population that requires policy and political attention as Tanzania moves towards universal health coverage. The authors have presented well the manuscript including the way the methodology, data analysis, and results have been presented. However, there are minor areas that I think the authors need to correct them to further improve the manuscript before its publication as follows:

Line 27: Delete "to" between among & CLWS.

Line 52: Write UN in its long form (United Nations) and then put the abbreviation UN in brackets.

Line 59: ADD "is" between It and estimated & ADD "are" between "up and to".

Line 64: Write HIV/AIDS in long form first and then put the abbreviation in brackets

Line 74: ADD "families” after poligamous and a semicolon before "also".

Line 76: ADD "in" between conducted and Tanzania.

Line 112: DELETE comma in the abbreviation CL,WS so that it appears as CLWS.

Line 120: DELETE "s" from the word “examines” so that the word will read examine & EDIT the CSO’s to be CSOs

Line 134: Authors need to consider deleting one among these two words (used employed) since they mean the same here.

Line 138: I suggest to authors to ADD "to have" between reported & around 39.6%

Line 142: I suggest to authors to REPLACE "cities" with “areas”.

Line 143: I suggest to authors to consider adding "which" between facilities and include OR change "include" to be including.

Line 148: ADD "of" between "provision & health".

Line 163: DELETE “Key Informants Interviews” and use only its abbreviation.

Line 167: DELETE “Focus Group Discussion” and use only its abbreviation

Line 479: DELETE Civil Society Organizations” and use only its abbreviation without brackets.

Line 500: DELETE comma after South so that it will read “South Africa”.

Line 508: CORRECT the spellings for Pakistani by DELETING “I” so that it will read Pakistan.

Line 531-533: The sentences (((The provision of medicines however is not based on medical laboratory diagnostic reports. However, monitoring of dosing has always been a challenge.)))) I suggest to authors to consider merging the two sentences by adding a semicolon instead of full stop after reports and DELETE "However", so that the sentence will read as follows: ""The provision of medicines however is not based on medical laboratory diagnostic reports; and monitoring of dosing has always been a challenge."""

Line 542: CHANGE “drives” to be “drivers”.

Line 543: The word “Consequences” to be a lower case (i.e., to start with small “c”.

Line 577: DELETE comma after South so that it will read “South Africa”.

Line 613: DELETE "s" in individuals so that it will read individual CSOs

Line 652: BACKSPACE "s" so that “clinic s” will read "clinics"

Line 658: REPLACE “Civil society organization” with its abbreviation CSOs

In Fig 1. Conceptual Framework on Health Care and Socio-Protection services among CLWS as summarized from different studies elsewhere:

I suggest to the authors to make corrections in Fig. 1 in the following areas

o On the DRIVERS OF CLWS: --Hunder (Food shortage) NEEDS to be corrected to "Hunger (Food shortage) &

o On "Enhancing Access to Health Care and socio-protection services" - the "Local Governmental" SHOULD BE EDITED TO BE "Local Government"

Reviewer #2: Dear Authors,

This contribution entitled:

"Challenges in accessing health care and socio-protection services among children living and working in streets in northwestern Tanzania: A qualitative study"

reveals important insight on health economic considerations for Tanzania's childhood socio-protection.

It is further applicable to the wider African and Global South context.

Yet in order to make these findings truly transnationally comparable evidence base cited should be diversified and expanded much beyond wealthy OECD countries - related finding.

This refers particularly to the developing LMICs health systems across the Global South particularly those that quite comparable to Tanzania and leading Emerging Markets.

For this purpose I warmly recommend to consider for inclusion a few of the below listed sources alongside with additional ones at authors own disposal:

https://globalizationandhealth.biomedcentral.com/articles/10.1186/s12992-020-00590-3

https://www.mdpi.com/1660-4601/16/17/3043

https://bmcmedicine.biomedcentral.com/articles/10.1186/s12916-022-02639-z

https://www.tandfonline.com/doi/full/10.1080/13696998.2023.2178164

https://www.mdpi.com/2071-1050/15/2/1372

erepository.uonbi.ac.ke/bitstream/handle/11295/161245/Mihajlo%20Jakovljevic.pdf?sequence=1

https://www.tandfonline.com/doi/full/10.1080/13696998.2022.2160608

https://resource-allocation.biomedcentral.com/articles/10.1186/s12962-022-00405-9

https://www.frontiersin.org/articles/10.3389/fpubh.2022.1062425/full?trk=public_post_feed-article-content

https://www.frontiersin.org/articles/10.3389/fpubh.2022.793648/full

https://www.hindawi.com/journals/bmri/2022/5514793/

https://health-policy-systems.biomedcentral.com/articles/10.1186/s12961-022-00822-5

https://pubmed.ncbi.nlm.nih.gov/30568938/

https://www.ncbi.nlm.nih.gov/pmc/articles/PMC9647898/

https://www.tandfonline.com/doi/full/10.2147/TCRM.S307587

https://pubmed.ncbi.nlm.nih.gov/34676266/

https://www.ncbi.nlm.nih.gov/pmc/articles/PMC9644335/

https://www.ncbi.nlm.nih.gov/pmc/articles/PMC9397501/

I remain willing to review the revised manuscript version.

Reviewer #3: The authors have conducted a phenomenological qualitative study of the barriers and opportunities for enhancing healthcare services and socio-protection for Children Living and Working on Streets (CLWS). The authors conducted Key Informants Interviews and Focus group Discussions that involved CLWS and workers in Civil Society Organizations. Data analysis revealed four themes: systems support, gender issues, economic support, and provision of education.

The followings are my comments:

In general, the manuscript needs a professional English editing service.

Abstract:

Line 27: “defiling” is not the right word here.

Line 36: what does “donor dependency” mean?

The introduction:

Lines 66-75: The introduction is lengthy, so I suggest removing this part that will render the intro more focused on CLWS in Tanzania.

Methods:

Line: 134: a reference to the phenomenological approach is needed.

Line 145: references that explain systematic random, purposive, and convenience sampling are needed.

Line 172: A reference for thematic content analysis is missing.

Line 180: a reference that describes the importance of investigator triangulation is needed.

Line 182: an in-text citation for NVivo 12 is missing.

Results:

216: The authors mentioned that five themes emerged but discussed only four themes and one subtheme.

The results section is too long. I recommend that the authors cut some of the non-essential verbatims of the included quotes.

Reviewer #4: Reviewer's comments on the manuscript. But generally, a lot of typo and grammatical errors. Title needs to provide a more comprehensive overview of the study's objective and focus. Objective is not clear, needs to be rephrased.

6. PLOS authors have the option to publish the peer review history of their article (what does this mean?). If published, this will include your full peer review and any attached files.

**Do you want your identity to be public for this peer review?** For information about this choice, including consent withdrawal, please see our Privacy Policy.

Reviewer #1: **Yes: **Eliudi Saria Eliakimu

Reviewer #2: No

Reviewer #3: **Yes: **Mohamed Hasan

Reviewer #4: **Yes: **Dr Chinedu Anthony Iwu

---

## [Decision Letter · Decision Letter 1]

24 Apr 2023

Challenges in accessing health care and socio-protection services among children living and working in streets in northwestern Tanzania: A qualitative study

PGPH-D-23-00312R1

Dear Elias,

We are pleased to inform you that your manuscript 'Challenges in accessing health care and socio-protection services among children living and working in streets in northwestern Tanzania: A qualitative study' has been provisionally accepted for publication in PLOS Global Public Health.

Best regards,

Collins Otieno Asweto, PhD

Academic Editor

Reviewer Comments (if any, and for reference):

Reviewer's Responses to Questions

**Comments to the Author**

1. If the authors have adequately addressed your comments raised in a previous round of review and you feel that this manuscript is now acceptable for publication, you may indicate that here to bypass the “Comments to the Author” section, enter your conflict of interest statement in the “Confidential to Editor” section, and submit your "Accept" recommendation.

Reviewer #1: All comments have been addressed

Reviewer #2: All comments have been addressed

Reviewer #3: All comments have been addressed

Reviewer #4: (No Response)

2. Does this manuscript meet PLOS Global Public Health’s publication criteria? Is the manuscript technically sound, and do the data support the conclusions? The manuscript must describe methodologically and ethically rigorous research with conclusions that are appropriately drawn based on the data presented.

Reviewer #1: Yes

Reviewer #2: Yes

Reviewer #3: Yes

Reviewer #4: (No Response)

3. Has the statistical analysis been performed appropriately and rigorously?

Reviewer #1: N/A

Reviewer #2: N/A

Reviewer #3: Yes

Reviewer #4: (No Response)

4. Have the authors made all data underlying the findings in their manuscript fully available (please refer to the Data Availability Statement at the start of the manuscript PDF file)?

Reviewer #1: Yes

Reviewer #2: Yes

Reviewer #3: Yes

Reviewer #4: (No Response)

5. Is the manuscript presented in an intelligible fashion and written in standard English?

Reviewer #1: Yes

Reviewer #2: Yes

Reviewer #3: Yes

Reviewer #4: (No Response)

6. Review Comments to the Author

Reviewer #1: (No Response)

Reviewer #2: Dear Authors,

Current manuscript revision is fully mature and acceptable for publishing without further changes.

I warmly endorse its acceptance.

best regards

Peer Reviewer

Reviewer #3: (No Response)

Reviewer #4: Thank you, there is an improvement from the original manuscript, however, certain fundamental issues which I tried to highlight through suggestions were not taken.

1. The title of any manuscript gives the first indication in a general way of what to expect. Title; “Challenges in accessing health care and socio-protection services among children living and working in streets in northwestern Tanzania: A qualitative study”

This title provides an expectation that the challenges which also includes the role of CSOs, will be explored in detail. But this was not the case, as the manuscript mainly focused on the role of CSOs as stated by you in your methods as follows; “A phenomenological approach was used to explore a full understanding of the individual, organizational, and social context factors on the role, prevailing barriers, and opportunities CSOs play in enhancing access to health care services and socio-protection among CLWS.”

And also, as evidenced from the results that you presented. This is what informed my initial suggest to recast your title, so that it limits our expectations according to what you have stated as your objectives. This was not considered.

2. Your objective was not clear, for example your stated objective was; “To explore the role of CSOs, preventing barriers and existing opportunities in enhancing access to health care and socio-protection services among to CLWS in Mwanza city, northwestern Tanzania.”

According to this objective, are you trying to explore the role of CSOs in preventing the barriers and creating existing opportunities towards enhancing socio-protection services among CLWS? If so, you need to rephrase, which I suggested initially. This was not considered.

However, if your intent was to explore separately the role of CSOs, barriers and opportunities then the statement in your methods on the phenomenological approach should be rephrased to be consistent with your objective.

3. Furthermore, on your statement “A phenomenological approach was used to explore a full understanding of the individual, organizational, and social context factors on the role, prevailing barriers, and opportunities CSOs play in enhancing access to health care services and socio-protection among CLWS,”

The above statement conveys the understanding that, the approach used, explored in depth the contextual factors (individual, organizational and social) that could influence the role of CSOs in preventing the barriers and creating opportunities that will enhance access to health care services and socio-protection among CLWS. If this is not your intent, then you have to rephrase

4. Finally, if your intent was to explore separately the role of CSOs, barriers and opportunities then your statements should be clear and consistent and also, your results should be organized clearly in line with the roles, barriers and opportunities after the initial demographics.

7. PLOS authors have the option to publish the peer review history of their article (what does this mean?). If published, this will include your full peer review and any attached files.

**Do you want your identity to be public for this peer review?** For information about this choice, including consent withdrawal, please see our Privacy Policy.

Reviewer #1: **Yes: **Eliudi Saria Eliakimu

Reviewer #2: No

Reviewer #3: **Yes: **Mohamed Mosaad Hasan

Reviewer #4: **Yes: **Dr Chinedu Anthony Iwu
